# The degree of internationalization of Chinese Multinationals along the belt and road initiative countries

**Olawoyin Gregory Adedigba** [ID] *⊙, **Runhui Lin**⊙, **Nizam Ud Din**⊙

Business School, Nankai University, Tianjin, China

⊙ These authors contributed equally to this work.
* olawoyingregory@gmail.com

**Data Availability Statement:** All relevant data are within the paper and its Supporting Information files.

**Funding:** This research was, in part supported by the National Natural Science Foundation of China

## Abstract

This study assesses the degree of internationalization of Chinese firms along the Belt and Road initiative countries. Most of the extant studies of the Belt and Road initiative have been qualitative, and where there have been quantitative studies, they have usually been at the aggregate level, and only a handful have used firm-level data to study initiative. Using a composite measure of the degree of internationalization, $DOI_{BRI}$, that composed of variables capturing the performance, structural and attitudinal dimensions of internationalization, comparative analysis of State-owned enterprises and privately owned enterprises turned up counter-intuitive results. Firstly, given that state ownership could be positively associated with the degree of internationalization of firms and because of the significance of the Belt and Road initiative, we expected the State-owned enterprises to dominate the $DOI_{BRI}$ rankings. We assessed the firms, and contrary to expectations, privately owned firms had a higher average degree of internationalization. Furthermore, we expected both state-owned enterprises and privately-owned enterprises to have similar levels of psychic dispersion. However, state-owned enterprises were more psychically dispersed. Suggesting that along the belt and road countries, the advantages of state ownership of Chinese multinationals may be attenuated.

## Introduction

The one belt one road initiative (interchangeably referred to as OBOR and BRI) is an all-encompassing policy initiative. It seeks to integrate countries along the ancient silk route and the new maritime silk route, through linkages, in five areas, namely; policy, trade, infrastructure, finance, and people [1]. This will be primarily achieved through massive infrastructure development projects in the 84 countries [2] that currently (October 2018) make up the BRI. Moreover, since 2013, $153 Billion has so far been invested by China in BRI countries [3].

There is a positive relationship between policy announcements and the internationalization activities of Chinesemultinational corporations(MNCs). Given China's unique institutional setting, major policy announcements have usually spawned, and increased outward flows of foreign direct investment [4] and policy announcements are one of the determinants of

research Grants (NSFC: 71772096, 71533002, and 71732005) and Major Projects of the Key Research Base of Humanities and Social Sciences of the Ministry of Education, China (16JJD630002).

**Competing interests:** The authors have declared that no competing interests exist.

Chinese outward foreign direct investment. In China, the government enjoys a significant influence on the economic landscape, especially with the presence of State-owned enterprises (SOEs) [5–7] and uses the SOEs to achieve specific policy outcomes [8]. It is logical, therefore, that the belt and road initiative, the most significant and most comprehensive policy initiative yet by the Chinese government, would generate a response by Chinese firms. Recent studies into the effect of belt and road initiative on Chinese foreign direct invest [6, 7, 9]support this.

However, despite the economic and political significance of the BRI, there are only a few empirical studies on the internationalization activities of Chinese multinationals in countries along the Belt and Road. The few available empirical studies have focused on the magnitude and flow of foreign direct investment into the countries along the Belt and Road initiative. While this gives a big picture view, it does not investigate internationalization per se, and there is a need for finer-grained analyses of the initiative and Chinese firms. In this study, using the BRI as the research context, we compute the degree of internationalization of Chinese firms in the BRI countries and then assess the relationship between ownership type and the degree of internationalization.

This paper aims to determine the degree of internationalization (DOI) of Chinese multinationals in the Belt and Road countries and to assess if state ownership translates to a higher degree of internationalization for Chinese multinationals. The diversity of the countries comprising the belt and road initiative means that virtually all of Dunning and Lundan's [10] conditions precedent to the internationalization of firms can be found within the Belt and Road initiative. These conditions *inter alia stipulate that a firm possessing ownership assets or advantages developed in the home country would seek to combine these with location or country-specific advantages(CSAs) through its ability to internalise missing or inefficient markets.* The belt and road initiative is essentially a large pool of CSAs considering the plurality of countries comprising the belt and road initiative. Chinese firms can unlock these provided they have the requisite combinatory nous to bundle them with their firm-specific advantages [11–13]. The Belt and Road initiative thus provides a tantalizing context to examine further Chinese multinationals in the light of extant internationalization theories.

Furthermore, while the importance and relevance of internationalization to the multinational firm is intuitively clear, its measurement has remained an issue of contention [14–18], with questions raised over the employment of single-item measures for a multidimensional concept such as internationalization [19]. In addition, because internationalization is multidimensional [20], its measurement is often context-specific, as such, there is no consensus amongst scholars concerning the choice of measurement of the degree of internationalization of firms [21]. Moreover, Sullivan decried this state of affairs claiming, "despite its theoretical and practical centrality, estimating the degree of internationalization (DOI) of a firm remains arbitrary" [19]. Meanwhile, Ietto-Gillies and London echo a similar sentiment albeit less critically, noting that there is no "acceptable" way to assess the degree of internationalization of firms [20]. Nevertheless, it is still essential to have a consistent measure of internationalization [14, 15].

Consequently, the degree of internationalization of Chinese firms in the Belt and Road countries, $DOI_{BRI}$is computed using a composite index based on Sullivan's [19] $DOI_{ints}$. He had highlighted the drawbacks of single-item measures of the degree of Internationalization prevalent in the literature and proposed his multi-item index as a useful alternative. Other researchers since then [22] have also employed the same index or slight variations thereof. Although, as far as we know, it is yet to be utilized with Chinese firms.

Our research contributes to both the literature on internationalization research in general and the internationalization of Chinese firms in particular by conducting a holistic assessment of the degree of internationalization of Chinese firms. Also, it contributes to the research into

the belt and road initiative by providing a firm level perspective on the initiative. In the next section, we introduce the study's hypotheses, following which we examine the literature and establish the theoretical foundation. The subsequent section covers the methodology and data analysis, after which we present the findings. The paper concludes with a discussion of findings and implications for theory and practice.

## Review of literature & theoretical framework

**Belt and Road initiative research.**    Research into the Belt and Road initiative spans various academic disciplines, including policy [23]. Economics [24] Management and International business [5–7]. However, given that the Belt and Road initiative was introduced in 2013, research into its influence and impact is still at infancy [6, 9]. Be that as it may, scholars across disciplines acknowledge and recognize the significance and potential of the initiative. Subject matter so far investigated in the context of the Belt and Road initiative include inter alia the role of the RMB along the one belt one road [25, 26], regional integration [27] as well as the impact of the initiative on Chinese outward foreign direct investment (OFDI) [6, 9]

One common feature of extant research is that most studies have been conducted at aggregate levels, that is, mainly at the regional, national, and industry level. These studies have mostly highlighted the outward foreign direct investment trends along the belt and road initiative countries, and only a handful of studies have examined the initiative from the firm's perspective. Also, only a few studies [6, 7, 9, 24] have been quantitative studies. While the question of whether the initiative will bring about an increase in China's outward foreign direct investment has been answered [6, 9], and there have been studies that breakdown the destinations and structure of Chinese outward foreign direct investment as well, such as the excellent work of Du and Zhang [7]. Under-investigated aspects of the literature include the actual firm level internationalization activities. However, this is somewhat of a surprise considering the scope and magnitude of the initiative. The countries that make up the initiative account for more than a third of both the world's GDP (22.9 trillion USD) and population (4.6Billion) [24].

Another theme in the literature is the industry distribution of the outward foreign direct investment. It is evident that compared to their private counterparts, State-owned enterprises dominate in the infrastructure projects along the Belt and Road initiative [7]. Suggesting that perhaps, State-owned enterprises are more willing to take on the significant financial burden and risks that accompany huge infrastructure projects.

Furthermore, because the quantitative studies have mainly examined the causal relationship between the BRI and changes in the levels of Chinese outward foreign direct investment, a preference for the difference in differences technique [6, 7, 9]was observed. These scholars have focused on determining the link between the launch of the BRI and changes in Chinese outward foreign direct investment patterns, and as there is a clear pre and post-event demarcation. That is, pre and post-2013, the year of the Belt and Road initiative's launch, the technique lends itself well to such an investigation. Their findings have mainly supported a positive relationship [6, 7, 9].

This study seeks to present the firm level perspective by investigating the internationalization activities of the Chinese firms in the belt and road countries. One way by which studying emerging markets multinationals can help extend theory is by providing the opportunity for understanding the development of firm capabilities and the economic and social conditions that make internationalization possible [28]. Considering the munificence of the location advantages in the belt and road initiative countries alluded to earlier, the Belt and Road initiative thus provides a unique context to observe the activities of Chinese Multinational firms.

**Internationalization of Chinese firms and the goldilocks debate.** Internationalization is the expansion of domestic firms into foreign markets through foreign direct investment [29]. Internationalization could be viewed as the process of increasing overseas' market commitments [30], and it is both a static and dynamic phenomenon. Theories of the multinational corporation have their roots in economic theory [31]. The theories have been classified based on the level of analysis vis a vis Macro-economic, Meso-economic and Micro-economic [29] while Forsgren [32] also distinguishes six main theories of the multinational firm based on the unit of analysis. Similarly, Rugman et al [13] note the evolution of internationalization theory from the country as the unit of analysis in the received theories of foreign direct investment such as the eclectic paradig, to the firm as the unit of analysis for instance in the case of internalization theory [12, 33, 34]as well as the internationalization process theory or the Uppsala model [30, 35]. While Kano and Verbeke [36] characterize New internalization theory as going further than the classical internalization theory in the analysis of the multinational firm by considering the multinational firm as a dynamic and multifaceted hierarchy and its unit of analysis is the subsidiary.

Apart from the unit of analysis, the theories also differ in the focus of analysis. For example, internalization theory is more concerned with how the firm deals with and responds to inefficiencies in external markets that raise transaction costs. The theories with the country as the unit of analysis usually have focused on determinants of foreign direct investment, from the home and host country perspectives. Whereas, the Uppsala model emphasizes the role of learning and experience [30] and in its updated version, insidership in the business networks of host countries [35, 37, 38]. Furthermore, while it could be argued that while the eclectic paradigm and internalization are concerned with why the firm will choose to engage in foreign direct investment, the Uppsala model attempts to answer how the firm implements the process. Narula [39] considers the eclectic paradigm, internalization theory, and the Uppsala model as the received theories of internationalization.

Internationalizing firms expand their business activities across national borders, usually through gradual increments in their commitments in foreign countries [30, 35], and in order to achieve this, they must possess certain advantages [40]. Which according to Dunning [11, 41–43], are ownership, location and internalization advantages (OLI) or firm-specific advantages(FSA) and country-specific advantages(CSA) as conceived by other scholars [12, 13, 44–46].Ownership advantages are those attributes or assets that confer competitive advantages on the firms. They include, for instance, a firm's proprietary knowledge as well as its processes and abilities to save costs. Location advantages may include natural endowments, large markets, and a supply of skilled labour. Internalization advantages have to do with the firms' ability to bypass market inefficiencies. While the foregoing can be considered the economic perspectives of internationalization, the Uppsala model emphasizes the behavioural dimension of internationalization through its focus on experiential learning as well as the liability of foreignness and outsidership, which attenuate internationalization.

Meanwhile, the internationalization of Chinese firms continues to generate debate in International business research as scholars are divided on the ability [44, 47, 48] or otherwise [49] of extant theories to explain the internationalization of Chinese firms. For instance, some scholars believe that Chinese firms have, by and large, conformed with the predictions of the eclectic paradigm [50, 51] while others doubt the ability of the eclectic paradigm to explain the internationalization of Chinese firms [52]. Reasons put forward for why extant theories seemingly struggle to explain Chinese multinational behaviour include neglecting the home country institutions [53] as well as neglecting the role of local owners of complementary assets [54].

In 2016, for the first time, China's outward foreign direct investments surpassed foreign direct investment inflows [55], becoming the second-largest source of outward foreign direct

investment after the United States. Consequently, with this transformation from a net foreign direct investment recipient to a net outward foreign direct investment destination, interest in the internationalization of Chinese firms has grown over the years [56]. There is a relative abundance of research into the motives and strategies of the internationalization of Chinese multinationals [44, 57–59], and the determinants of Chinese outward foreign direct investments include, among other things, market size, institutional settings, and government policy [4]. While the motives for internationalization of Chinese firms are *inter alia* asset seeking, and escape from inefficient home- country institutions or "institutional arbitrage" [4, 56, 57, 60–62].

However, Rugman et al. argue that the successes of the internationalization of Chinese firms have been blown out of proportion [44] as their performance lags when compared with western multinationals in the same industry. Collinson and Rugman [45] also contend that Chinese Multinationals are regional and, as such, cannot be accurately described as engaging in internationalization. Nevertheless, research into the internationalization of Chinese firms has been vibrant mainly because Chinese firms are relatively unique [53], and studying them allows the testing of theories mainly developed with western multinationals in mind [50]. Secondly, the relative speed with which Chinese multinationals have successfully internationalized flies in the face of received wisdom and has prompted calls for the development of new theories for emerging Multinationals [49], or at the very least extensions of extant theories. For instance, Berning&Holtbrügge [60] assert that existing theories need to be extended or modified in order to be applicable in the Chinese context. However, others believe that extant theories have sufficient explanatory power [44, 48, 63]. Whether for, or against extant theories, it is evident that the internationalization of Chinese firms as a field of study provides a unique context for management and business research and has the potential to enrich international business and management studies.

Furthermore, given that new policy announcements encourage the flow of outward foreign direct investment (OFDI) [4, 64], we can, therefore, expect that the launch of the Belt and Road initiative would precipitate an increase in OFDI and the internationalization of Chinese firms. Indeed early assessments of the Belt and Road initiative have found empirical support for increased outward foreign direct investment [7, 9]. However, as stated earlier, while these studies have painted the overall picture of Chinese capital flows, the individual firms and the transactions behind these flows remain unexplored. This paper aims to fill that gap.

**Hypotheses.** Since Chinese multinationals typically fall under two categories; State-owned enterprises (SOEs), and privately owned Enterprises (POEs) and because they typically have different corporate structures, they contend with different operational constraints and the amount [65] of resources available to them. Consequently, these structural and capacity differences mean that both types of firms usually approach internationalization differently [65]. Will the ownership structure of Chinese multinationals influence their response to the belt and road initiative and shape their internationalization efforts along the Belt and Road countries? Or is ownership structure likely to be irrelevant to the internationalization of Chinese firms along the belt and road countries? Since government policy is a strong determinant of Chinese foreign direct investment [4], we expect a positive relationship between the internationalization of Chinese firms and the expansion of the Belt and Road initiative. Furthermore, because of their unique structure, SOEs are characterized by a close working relationship with the government and because the government exerts influence on the degree of globalization of SOEs through ownership [66] and SOEs are required to fulfil national objectives [67], we propose;

**H1**: State-owned enterprises would have a higher $DOI_{BRI}$ than privately owned enterprises

One of the principal challenges of a firm entering into international markets is how to overcome the liability of foreignness, that is, the extra costs of operating in new markets brought about by the psychic distance between the home country of the firm and the host country [13, 30]. Psychic distance encompasses differences in cultures, norms, and values [68]. The Belt and Road initiative spans more than 60 countries across all continents; it stands to reason, therefore, that this diversity may increase the psychic distance, which would be inimical to successful internationalization. At the same time, Buckley et al. [4] show that geographical distance is also a significant determinant of foreign investment outflow, and Du and Zhang [7] have shown that a significant percentage of outward foreign direct investment into the belt and road initiative countries are to East and Central Asian countries. We conceive that because Chinese SOEs and POEs originate from the same home country and share the same cultural attributes, consequently, they will contend with similar levels of psychic distance. We, therefore, propose that

**H2**: Along the Belt and Road countries, psychic dispersion of State-owned enterprises and privately owned enterprises would be identical

## Materials and methods

### Sample and data

The research employs a linear combination of the dimensions of internationalization in developing its index to measure the degree of internationalization. Because the decision to internationalize is inherently a firm-level decision, we are interested in not just the direction and magnitude of outward direct investment. Therefore, our investigation is best served by the use of firm-level data. Given that the use of firm-level data in the research of Chinese outward foreign direct investment (OFDI) provides the opportunity for more in-depth analysis [46, 68]. We collected data from multiple sources, including databases like Heritage foundation's China Global investment tracker, WIND, and CSMAR. These three databases provide information that runs the gamut of foreign direct investment flows, providing both mergers and acquisition as well as Greenfield data. For instance, the China Global investment tracker provides data on all cross border deals concluded by Chinese firms worldwide that surpass the US $100 million threshold and contains data for as far back as 2005. The data includes both direct investment transactions and contracts. It also specifies which transactions are Greenfield investments. Most importantly, the data distinguishes whether a recipient country belongs to the Belt and Road initiative or not. The use of multiple data sets allows for the triangulation of Data, and therefore it improves the quality of data used in the analysis.

While some scholars have relied on the use of single-item measurements, usually the ratio of foreign sales to total sales [44, 69], others have preferred a multi-item approach [14, 22, 70]. However, the view that composite measures are better suited to the measurement of the degree of internationalization is widely held [14, 19, 22, 70, 71]. Nevertheless, Ramawamy et al. [72], in a critique of Sullivan [19] observed that composite measures might not necessarily improve the understanding of internationalization. However, Sullivan [73] maintains the usefulness of a multi-item index given it adequately addresses issues of construct validity and item validity. Multi-Item Indices allow for the inclusion of non-performance variables when assessing internationalization. Given that multi-item measures allow for a fine-grained assessment of internationalization, we follow prior scholars [14, 19, 66, 71] in adopting a composite measure of the degree of internationalization countries.

The first step in sample generation was to extract from the global investment tracker all the transactions completed by Chinese firms from 2013 till 2018. To be included, in addition to

being from 2013, it must have occurred in a Belt and Road country. A shortlist of transactions satisfying these two conditions was compiled. The originating firms of these transactions were then identified and collated. This approach ensured that there was an even spread and allowed for a mitigation of the selection bias [44, 56], which would have occurred had we selected the companies from a pre-existing list, e.g., top 100 Chinese firms. There is still some form of selection bias, however. As the China Global investment tracker only includes transactions of a minimum of USD 100million. That means companies who have investments below that threshold will a priori not make it into our list. The global investment tracker database classifies transactions based on whether the deal was an investment or a construction contract. Using the rubric above, we had 163 unique firms that had been involved in investment transactions and 94 firms that primarily had construction transactions giving a total of 257 firms. The shortlist is summarized in Table 1 below.

In the second stage, only firms with publicly available data were retained. These were composed primarily of listed firms and firms like Huawei that, although not listed, have publically available data. This was necessitated by data availability challenges, as listed firms have disclosure requirements; therefore, the necessary data is readily available. The data of the short-listed firms were then sourced from the WIND database as well as CSMAR in addition to the published annual results. These include figures for foreign and total sales, as well as overseas, and foreign subsidiaries.

Finally, we applied the constraint that firms must have significant operations or activities in overseas markets. Internationalization, as a concept, presupposes the initiation of and participation in cross border business activities. In any case, Rugman et al. [44] Consider the multinational firm as one with at least ten percent of sales in foreign markets and with at least three foreign subsidiaries. The multinational firm is also one that engages in value addition activities across borders [10] as well as one that organizes the linkages between employees located in more than one country through employment contracts [52]. At the same time, Fitzgerald & Rowley [74] define it as a firm that has significant investments and large scale business activities in different countries while maintaining ownership and control. Considering these attributes of multinationals, our definition is not as weighted towards the proportion of foreign sales as an indicator of multi-nationality *a la* Rugman. For instance, Alibaba reported no foreign sales for the period under review. However, it had 30 foreign subsidiaries, which indicates a significant investment and involvement in overseas markets, albeit one suggestive of a strategic rather than commercial nature. Were we to apply the 10% of the total sales requirement, we would have had to exclude Alibaba from the sample, which would be odd given the evidence of its international involvement. This underscores Sullivan's [19] point that the use of foreign sales ratio is not a sufficient measure of internationalization. In any case, the internationalization motives of Chinese firms vary, from the purely market seeking to the strategic asset seeking motive [10] to the strategic intent perspective [75]. Therefore to be included in

**Table 1. Chinese deals along BRI countries from 2013–2018.**

| Item | Investment | Contracts |
|------|-----------|-----------|
| No of deals 2013–2018 | 248 | 527 |
| No of initiating companies | 163 | 94 |
| Dollar value | $153 billion | $267 billion |
| No of countries | 46 | 57 |
| Regions | 6 | 7 |

Source: China global investment tracker

our sample, firms must have engaged in foreign direct investment along the Belt and Road countries in addition to either having a foreign subsidiary and or have sales from overseas markets. This conceptualization of multi-nationality is broadly in agreement with Dunning's definition of the multinational as "an enterprise that engages in foreign direct investment (FDI) and owns or, in some way, controls value-added activities in more than one country" [10]

With this final criterion, our sample was reduced to 92 firms. A major casualty of this stage was Jingdong, the Chinese e-commerce giant, with neither foreign sales nor foreign subsidiaries.

## Variables

Internationalization is a multi-dimensional concept, as such a wide array of variables, can be used to operationalize its measurement [19, 71]. There is a location dichotomy inherent to the concept of internationalization, that is, what goes on in one location versus what goes on outside of it [71]. Generally, the home country of the multinational firm on one hand and foreign territories on the other hand. Consequently, in developing measures for the degree of internationalization, scholars have utilized ratios that capture the percentage of foreign-initiated or oriented activities to the total activities of the firm, including the ratio of foreign sales to total sales [44, 69, 76]and foreign assets to total assets [8]. In addition to the location dimension mentioned earlier, other key dimensions identified in the literature include structural, attitudinal, and Performance [19], intensity, geographic-scope extensity, and geographic-scope concentration [20] as well as breadth and depth scope [16].

Sullivan's [19] $DOI_{INTS}$ was a five item composite index made up of a linear combination of ratios. The ratios were foreign sales and total sales, foreign assets, and total assets, top managers' international experience, and psychic dispersion of operations. We use a modified version of the $DOI_{INTS}$ for this study. Owing to data availability constraints, the foreign asset to total assets ratio was omitted as was the top managers' international experience, while including a measure for geographical dispersion a la Ietto-Gillies [71]. We use a four-item index, as shown below. The degree of internationalization is computed for each shortlisted firm by plugging the ratios for each firm into the equation below. In adopting this approach, we agree with Dörrenbächer [77] that Sullivan's measure is one of the few that combines the main dimensions of Internationalization into one measure.

$$DOI = FSTS + OSTS + PDIO + GD \qquad (1)$$

The main dimensions of internationalization identified in the literature include; performance, structural, and attitudinal dimensions [19, 77], intensity, and extensity dimensions [20] as well as real and financial [21]. We do find, however, that the dimensions and the variables to operationalize them often overlap. For instance, while Sullivan [19] considers foreign sales to total sales ratio as a performance variable, Ietto-Gilles [71] considers it a measure of intensity, the value of activities per location. This difference, in our view, is semantic, as they still capture the same effect. Consequently, we ensured the key dimensions of internationalization were captured by our measure. For instance, the omission of the assets ratio, which expressed the structural dimension in Sullivan [19], does not significantly impact the usefulness of the index as the structural dimension is still aptly captured by the subsidiary ratio. While the sales ratio still captures the performance dimension and psychic dispersion operationalize the attitudinal dimension. In essence, we have one ratio to capture each dimension, as identified by Sullivan [19]. Furthermore, given that compounded distance mitigates the internationalization efforts of firms [13] and just assessing the volume of overseas activities

does not say much about how dispersed or extensive those activities are [14, 71], a measure of the geographic dispersion of the firms was included in determining how widespread the firm's activities are along the Belt and Road Countries. The average of the three years 2015,2016,2017 were used to compute foreign sales to total sales ratio, while the value of overseas subsidiaries to total subsidiaries was determined using single-year data, in this case, the year 2017.

## Performance/Intensity (FSTS)

The ratio of foreign sales to total sales is the most utilized measure of the degree of internationalization [46], and several studies have solely utilized this variable to measure the degree of internationalization [69, 77]. While it is indeed a good proxy, it has flaws because it is inherently biased towards large firms and firms with an inordinate amount of foreign sales that might originate in a single market, for instance [15]. Moreover, as observed by Ietto-Gillies and London [20], the foreign sales to total sales ratio mainly focuses on the home vs. foreign dichotomy without providing insights dispersion. While it adequately captures the performance dimension of internationalization, it fails to capture the other dimensions of internationalization. For example, if we take two multi-national companies X and Y, X with foreign sales of $10million and Y with $5 million, where X's foreign sales come from two countries, and Y from 5 territories. If both had total sales of $20 million, and we compute their foreign sales to total sales ratio, that measure will show X as having a higher degree of internationalization. This approach would completely ignore the associated complexity of operating in more geographically diverse and potentially psychically diverse markets. This is why the FSTS alone does not paint a robust picture of the internationalization activities of firms.

FSTS captures the quantum of business done by the multinational and what proportion of its performance comes from international markets. It is a good first-order measure of international involvement of the firm. If a firm is a strategic asset seeking international firm, the FSTS measure may not adequately capture this aspect of internationalization. We argue that it focuses too narrowly on the market seeking motive of internationalization; therefore, it is better to use it as part of a multivariate index rather than a stand-alone measure.

## Structure (OSTS)

Following Sullivan [19, 73], we use overseas subsidiaries to total subsidiaries ratio to capture the structural dimension of internationalization. Since Chinese firms also base their internationalization on rational analysis [75] the types and locations of Chinese multinationals' subsidiaries may reflect their strategic intent given subsidiaries vary in role and type [78], e.g., overseas subsidiaries that are not directly involved in revenue-generating activities may focus on research and development, administration or even become the so-called center of excellence [79]. Subsidiaries located in tax havens may contribute immensely to the bottom line of the firm even though they might not generate revenues from sales. Moreover, as highlighted by Hassel et al. [21], their investigation of financial internationalization of German firms revealed that extant studies have often focused on the production or real dimension of internationalization to the exclusion of governance dimensions of internationalization. We acknowledge the non-financial dimensions of internationalization by considering the ratio of overseas subsidiaries to total subsidiaries not just as a purely structural measure, but as an indicator of strategic intentas well.

## Extensity/Psychic Dispersion of International Operations (PDIO)

The liability of foreignness that multinationals face is a key determinant of the success of their internationalization process [13, 35, 80], and psychic distance influences the liability of

foreignness [30]. The variable PDIO measures the psychic dispersion of the firm's operations. That is, how psychically diverse the firms' operations are, along the belt and road initiative countries. Sullivan [19] measured psychic dispersion using Ronen and Shenkar's [81] psychic maps. It is possible to have territories that might be geographically proximate but psychically or culturally distant and vice versa, where countries are psychically proximate despite the geographical distance therefore, it is useful to measure psychic dispersion. An easy example would be Great Britain and the USA's cultural proximity, even though they are separated by a large geographical distance.

Although Sousa & Bradley [68] make a distinction between psychic and cultural distance, we find either conceptualization sufficient for our study because they both relate to how distinctive attitudinal traits compare and vary across locations. Having said that, however, it became quickly apparent that Ronen and Shenkar's [81] classification would not suffice for this study. Ramaswamy et al. [72] already pointed out that the country clusters did not sufficiently address the diversity in the world, we share their view that the 'independent' group does not really provide any useful insight. For example, using this study's sample of the belt and road countries, the Ronen and Shenkar [81] classification considers both Nigeria and Croatia as "independent", and one would be hard-pressed to argue that these two countries have a similar psychic makeup. The Global investment tracker's classification also proved inadequate, as their classification is more geographically oriented rather than psychological or cultural. We modified Ronen and Shenkar's [81] classification to better suit the study by including a tenth cluster–Sub-Saharan Africa. We find this more relevant to the study as a significant number of belt and road countries are found here. Moreover, there is more to be said for the psychic similarity of Nigeria and Cameroon than for Nigeria's and Croatia. Table 2 shows the classification.

## Scope/Geographic dispersion

The uniqueness of the Belt and Road initiative is that it encompasses many countries; therefore, it is useful to identify how widespread the activities of Chinese multinationals are, along the belt and road initiative. Consequently, a fourth item was included in our index; it is intended to capture the *Geographic* extensity of the firms along the Belt and Road initiative countries, given the relevance of geographic diversity [20] in this context. We follow Ietto-Gilles [71] in using a count of the number of countries along the Belt and Road initiative countries in which the firm has activities, normalized by the total number of belt and road countries which stood at eighty-four [2] at the time of this paper. Furthermore, Ramaswamy et al. [72] argued in fact that a simple count of the number of countries might be a better measure of the dispersion of internationalization of Multinational firms, and this study adopts it as our measure of geographical dispersion. Below, Table 3 presents a description of the variables and data sources.

The ratio of foreign sales to total sales (FSTS) and overseas subsidiaries to total subsidiaries (OSTS) correspond to both Ietto-Gilles' [71] intensity dimension and Sullivan's [19] performance dimension. PDIO captures the attitudinal dimension, as identified by Sullivan [19], while the last variable GD pertains to the geographic extensity identified in Ietto- Gillies [71].

## Results

Descriptive statistics for the State-owned enterprises (SOEs) indicate that the same firms had the Maximum and minimum scores in three of the variables. In this case, Qingdao Heng-shunZhongshen had the Maximum score in PDIO, OSTS, and GD measures. While Yunan Energy and Shanghai Shentong were tied for the lowest scores in all three variables. Qingdao

**Table 2. The countries of the BRI mapped on a modified version of Ronen and Shenkar's Psychic Zones that includes Sub-Saharan Africa.**

| Nordic | Germanic | Anglo | Latin Europe | Latin America | Far Eastern | Arab | Near Eastern | Independent | Sub Saharan |
|---|---|---|---|---|---|---|---|---|---|
| | Austria | New Zealand | Portugal | Bolivia | Bangladesh | Algeria | Afghanistan | Antigua and Barbuda | Angola |
| | | | | Chile | Brunei | Bahrain | Albania | Dominica | Burundi |
| | | | | Costa Rica | Cambodia | Egypt | Azerbaijan | Fiji | Cameroon |
| | | | | Dominican Republic | India | Iraq | Belarus | Grenada | Cape Verde |
| | | | | El Salvador | Indonesia | Jordan | Bhutan | Guyana | Chad |
| | | | | Uruguay | Laos | Kuwait | Bosnia and | Herzegovina | Congo |
| | | | | Venezuela | Malaysia | Lebanon | Bulgaria | Hungary | Cote d'Ivoire |
| | | | | | Myanmar | Libya | Croatia | Israel | Djibouti |
| | | | | | Nepal | Morocco | Czech Republic | Lithuania | Ethiopia |
| | | | | | Pakistan | Oman | Estonia | Malta | Gabon |
| | | | | | Philippines | Qatar | Georgia | Mauritania | Gambia |
| | | | | | Republic of Korea | Saudi Arabia | Greece | Moldova | Ghana |
| | | | | | Singapore | Syria | Iran | Mongolia | Guinea |
| | | | | | Sri Lanka | Tunisia | Kazakhstan | Montenegro | Kenya |
| | | | | | Thailand | Turkey | Kyrgyzstan | Niue | Madagascar |
| | | | | | Timor-Leste | United Arab Emirates | Latvia | Panama | Maldives |
| | | | | | Viet Nam | Yemen | Macedonia | Papua New Guinea | Mozambique |
| | | | | | | | Palestine | Poland | Namibia |
| | | | | | | | Russia | Romania | Nigeria |
| | | | | | | | Serbia | Samoa | Rwanda |
| | | | | | | | Tajikistan | Slovakia | Senegal |
| | | | | | | | Turkmenistan | Slovenia | Seychelles |
| | | | | | | | Uzbekistan | Suriname | Sierra Leone |
| | | | | | | | | Trinidad and Tobago | Somalia |
| | | | | | | | | Ukraine | South Africa |
| | | | | | | | | | South Sudan |
| | | | | | | | | | Sudan |
| | | | | | | | | | Tanzania |
| | | | | | | | | | Togo |
| | | | | | | | | | Zambia |
| | | | | | | | | | Zimbabwe |

Zhongshen is a conglomerate involved in power projects as well as infrastructure development projects along the belt and road initiative. It is a relatively young company established in 1998. Meanwhile, Yunnan Energy Investment Company is also a company involved in the power sector and is relatively younger, having been founded in 2012. The second company, tied with the lowest values, is Shanghai Shentong, a container terminal company founded in 2005. It is unsurprising to note that these companies are all involved in the Infrastructure sector, as this is one of the major thrusts of the Belt and Road Initiative. Table 4 presents the descriptive statistics of the State-owned enterprises and Table 5 statistics for the privately-owned enterprises.

On the other hand, the privately owned enterprises (POEs) showed more variety in the companies with the maximum values for the variables, unlike the State-owned enterprises

**Table 3. Summary of variables.**

| Variable name | Description | Source |
|---|---|---|
| DOI | A multi-item index that measures the level of international involvement of Chinese MNCs. It is a linear addition of 4 variables FSTS+OSTS+PDIO+GD | WIND, CSMAR |
| FSTS | Ratio of foreign sales to total sales. Calculated as 3 year AVG Foreign sales/Total sales for 2015–2017 | WIND, CSMAR |
| OSTS | Ratio of overseas subsidiaries to total subsidiaries. Calculated as Overseas Sub/Total Sub. | WIND, CSMAR |
| PDIO | A measure of the degree to which the operations of the firms are *psychically* Dispersed along the Belt and Road Initiative. Computed as the ratio of psychic zones where a firm is operational/total psychic zones along the BRI | WIND, CSMAR |
| GD | A measure of the degree to which the operations of the firms are *geographically* dispersed along the Belt and Road Initiative. Computed as the ratio of belt and road countries where a firm is operational/total number of belt and road countries | Official OBOR Website |

(SOEs). For example, three different companies posted the maximum values for GD, PDIO, OSTS, and FSTS. They are Sino great wall, Huawei, and Qinjian, respectively. However, similar to the SOEs, the companies that posted the minimum values stayed consistent. In this case, three of the variables had the same companies, Zhongrun and Jumei both had the same value for the GD, OSTS, and FSTS variables. Although they had the lowest values for the PDIO variable as well, they share that distinction with 38 other companies. Also notable is the fact that the POEs have a wider range of industry represented amongst the maximum scoring firms. For instance, Huawei is a telecommunications company, while Sino great wall is a construction and engineering company, and lastly, Qinjian is a construction company as well. It is useful to note that Qingjian scored the maximum value on the overseas to total subsidiaries ratio on account of its three subsidiaries being overseas subsidiaries. Fig 1 displays the sectoral composition of all the investments made by the firms, while Figs 2 and 3 present the breakdown by sectors for the privately-owned and State-owned enterprises respectively. Notably, the POEs with the highest scores are older than their SOEs counterpart by some distance. For example, Qingjian was founded in 1952, while Sino great wall and Huawei were founded in 2001 and 1986, respectively. S1 Appendix provides the firms' ages.

The average DOIBRI of the sample was 0.67, and when broken down into State-owned enterprises and privately owned enterprises, we have 0.59 and 0.75 respectively. Table 6 presents the descriptive statistics showing the POEs vs. SOEs and Table 7 overall descriptive statistics. Compared to state-owned enterprises the privately owned enterprises posted higher mean DOIBRI values. As a matter of fact, only two State-owned enterprises were in the top ten as ranked by the DOIBRI values. Qingjjian, a private firm, recorded the highest DOIBRI. Consequently, H1 is unsupported. On the psychic dispersion measure, State-owned enterprises had an average of 0.18, while for private enterprises, the value was 0.11, indicating that the State-

**Table 4. Descriptive statistics SOEs.**

| Variable | OBS | Mean | Std.DEV | Min | Max |
|---|---|---|---|---|---|
| FSTS | 47 | 0.169 | 0.196 | 0 | 0.974 |
| OSTS | 47 | 0.175 | 0.244 | 0 | 1 |
| PDIO | 47 | 0.189 | 0.125 | 0.1 | 0.5 |
| GS | 47 | 0.060 | 0.091 | 0.011 | 0.404 |
| $DOI_{BRI}$ | 47 | 0.594 | 0.383 | 0.111 | 1.808 |

**Table 5. Descriptive statistics POEs.**

| Variable | OBS | Mean | | Std. Dev. | Min | Max |
|---|---|---|---|---|---|---|
| FSTS | 44 | 0.29 | | 0.3 | 0 | 1 |
| OSTS | 44 | 0.32 | | 0.32 | 0 | 1 |
| PDIO | 44 | 0.12 | | 0.05 | 0.1 | 0.3 |
| GD | 44 | 0.02 | | 0.02 | 0.01 | 0.1 |
| DOIBRI | 44 | 0.75 | | 0.51 | 0.11 | 2.11 |

owned enterprises scored higher on that measure; therefore, H2 is unsupported. If we looked at the dispersion measures alone, the SOEs indeed ranked higher than the POEs; this suggests that perhaps state-owned enterprises are unfazed by and better equipped to deal with the challenges of operating in more psychically and geographically dispersed locations. It may also be that state ownership has less influence on the internationalization of Chinese firms along the Belt and Road initiative, and other moderator variables are better associated. Liu et al [82], for instance, found market potential to be a strong determinant of Chinese outward foreign direct investment along the Belt and Road countries. S2 Appendix presents a list of Belt and Road countries where the firms have invested and the corresponding number of Psychic Zones.

The rankings of the other dimensions revealed that State-owned enterprises ranked higher in the extensity dimensions, that is, the psychic and geographic dispersion. Whereas, the private firms ranked higher in the intensity and performance measures, foreign sales to total sales (FSTS) and overseas subsidiaries to total subsidiaries (OSTS). Overall, even though the number of firms in the sample was identical between state and privately ownedfirms (47 and 44 respectively), we discovered that the firms with the highest degree of internationalization scores were privately owned enterprises and not State-owned enterprises. This was unexpected, given the relative financial might of State-owned enterprises and typical preferential treatment they enjoy, the expectation was that they would be at the vanguard of the internationalization rankings along the belt and road countries. It should be noted, however, that in terms of sheer numbers of transactions initiated, the state-owned firms still rule the roost. It is only when this is normalized do we see the result discussed above. This is illustrative of the usefulness of a composite index in computing the degree of internationalization, as it discounts firm size. We also found that Alibaba has a preponderance of its subsidiaries abroad, with approximately

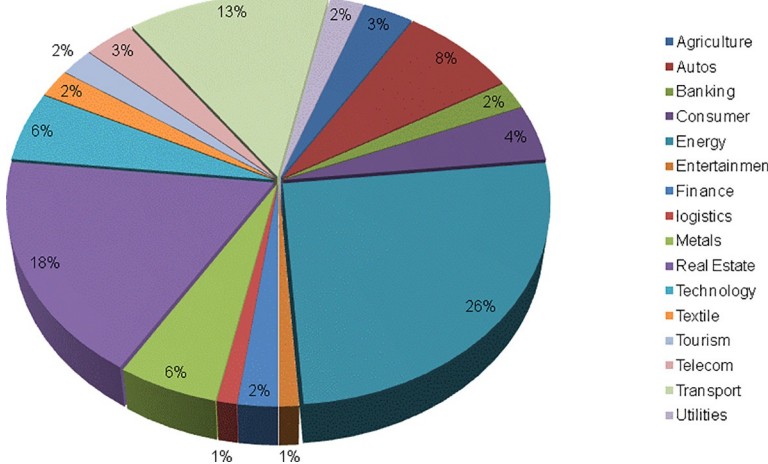

**Fig 1. Overall sectoral composition.**

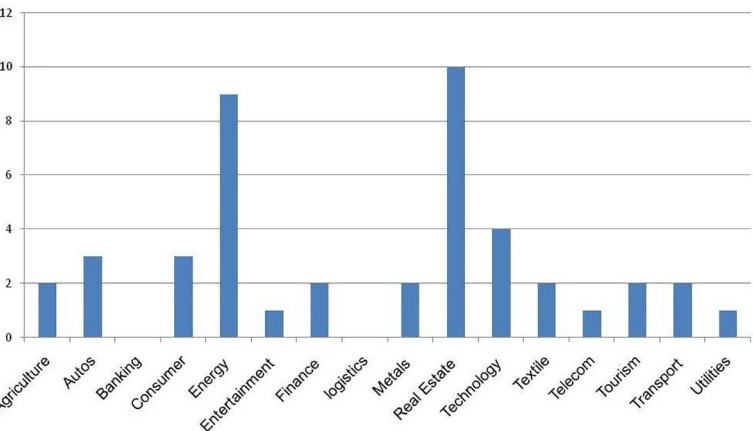

**Fig 2. Sector breakdown of privately owned firms.**

70% of its total subsidiaries located overseas. This, however, contrasts with its foreign to total sales ratio, where no figures for foreign sales were recorded in the annual reports. S1 Appendix presents the score of each firm for each of the variables, and S2 Appendix the list of countries where the firms.

The correlation matrix, presented in Table 8 below, indicates a strong and positive correlation between the dispersion variables psychic dispersion of international operations(PDIO) and geographic dispersion (GD) at 0.85. Cronbach's alpha of the index was 0.4889. These measures are however, negatively correlated with both foreign sales to total sales (FSTS) and overseas subsidiaries to total subsidiaries (OSTS). This is in line with the prediction of the Uppsala model [30, 35] that firms would expand into psychically and geographically proximate locations before venturing further afield.

## Discussion

Data availability has remained a bane to the measurement of degree of internationalization of firms [77], and as articulated by several scholars [14, 19, 20, 73, 77], researchers are constrained

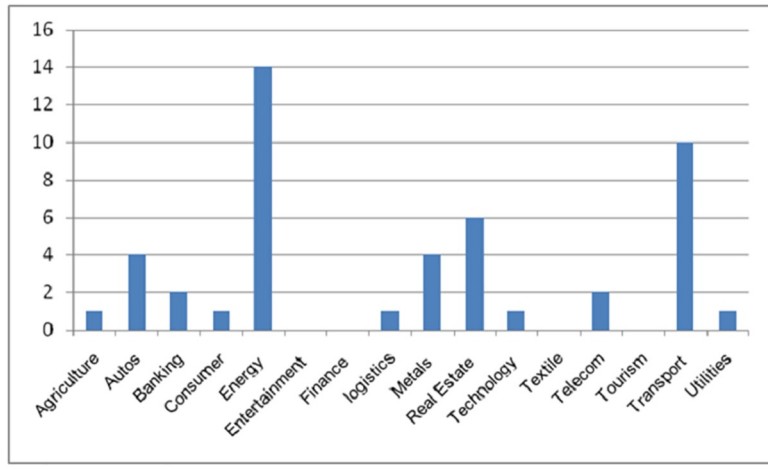

**Fig 3. Sector breakdown of privately owned firms.**

**Table 6. Descriptive statistics POEs vs SOEs.**

| Variable | Combined mean | SOEs (AVG) | POEs(AVG) |
|---|---|---|---|
| FSTS | 0.2280 | 0.1691 | 0.2908 |
| OSTS | 0.2461 | 0.1753 | 0.3218 |
| PDIO | 0.1549 | 0.1893 | 0.1181 |
| GD | 0.0406 | 0.0605 | 0.0194 |
| $DOI_{BRI}$ | 0.6698 | 0.5944 | 0.7503 |

by available and accessible data. This study suffers the same fate, as we had to omit certain variables owing to data unavailability/inaccessibility. For example, the top management international experience. While we acknowledge that alternative methods or data may exist, our study was carried out within this constraint.

Regarding items comprising the index, computed Cronbach's alpha of 0.48 barely lies within acceptable limits of 0.7 for exploratory research that is widely cited in the literature [19, 22]. However, the usefulness of the alpha in measuring validity and reliability has been questioned [83]. Cronbach's alpha is influenced by factors that may not reflect the reliability of a scale [84] as well as by the number of items in the scale and the sample size; therefore our limited sample size is the likely source of the low value rather than the unsuitability of the measure. In any case, Panayides [85] advises caution when reporting alpha values; as higher values do not necessarily indicate higher reliability. Moreover, ex-ante, alpha is neither capable of measuring uni-dimensionality nor reliability [86]. Given that the goal of the $DOI_{Bri}$ was to capture the multidimensional nature of internationalization and higher alpha values could be associated with a narrow coverage of a construct [85] our alpha value does not debase the utility of the index.

Furthermore, our finding that the State-owned enterprises did not have higher $DOI_{Bri}$ values as predicted could be down to the fact that given the development of Chinese multinationals over the last decade, private firms have been able to amass formidable resources and developed capabilities that neutralize the advantages accrued by State-owned enterprises. It is also possible that because the advantages enjoyed by State-owned enterprises were borne of the relatively weak institutions and market inefficiencies in existence, and because there is a relationship between reforms and the degree of internationalization [66], recent reforms have been successful to such an extent that they have degraded the advantages of State-owned firms over privately owned enterprises. Lending credence to the view that under certain conditions, the behaviours of State-owned enterprises and privately owned firms, in terms of international activities converge and become indistinguishable [8, 66], which suggests that perhaps along the Belt and Road initiative, the advantages bestowed by state ownership are irrelevant to successful internationalization. Although the institutional arbitrage [87] perspective is that privately owned enterprises may be escaping disadvantageous positions in the home market, the fact that there is not such a wide variance between the mean $DOI_{Bri}$ of the State-owned

**Table 7. Overall descriptive statistics.**

| Variable | Observations | Mean | Std. Dev. | Min | Max |
|---|---|---|---|---|---|
| $DOI_{BRI}$ | 91 | 0.6698 | 0.4515 | 0.1119 | 2.1108 |
| FSTS | 90 | 0.2305 | 0.2603 | 0 | 0.9989 |
| OSTS | 91 | 0.2461 | 0.2911 | 0 | 1 |
| PDIO | 91 | 0.1549 | 0.1035 | 0.1 | 0.5 |
| GD | 91 | 0.0406 | 0.0697 | 0.0119 | 0.4047 |

**Table 8. Correlation matrix.**

|  | DOI | FSTS | OSTS | PDIO | GD |
|---|---|---|---|---|---|
| DOI | 1 | | | | |
| FSTS | 0.7701 | 1 | | | |
| OSTS | 0.7721 | 0.3685 | 1 | | |
| PDIO | 0.1788 | -0.1008 | -0.2 | 1 | |
| GD | 0.1588 | -0.1012 | -0.21 | 0.855 | 1 |

enterprises and the privately owned firms, suggests that at least within the Belt and Road initiative context, this may not be the case. However, these explanations may only be relevant in the context of the Belt and Road initiative because being the brainchild of the government, deliberate measures may have been taken to reduce the red tape that would have adversely affected internationalization efforts of private firms. Future research could test this by comparing the activities of privately owned Chinese firms in Belt and Road to non-Belt and Road initiative countries.

Secondly, we find that State-owned enterprises scored higher on our measure of psychic dispersion, contrary to our prediction of the opposite, which was indicative of more psychically diverse and dispersed operations. It is possible that they are better equipped to deal with the liabilities of foreignness that goes with operating in psychically distant countries. It could be, for instance, that since State-owned enterprises usually prefer government to government relationships [65], they are better able to get host countries' governments to safeguard their investments, which in turn mitigates the liability of foreignness. The fact that State-owned enterprises have a preponderance of their activities in the infrastructure development projects in the Belt and Road countries [7] lends credence to this, as these projects are likely to be negotiated, agreed and commissioned at the highest levels of the host country governments. This suggests that state ownership may not matter much to the overall degree of internationalization, but that it is crucial in overcoming the liability of foreignness.

Firm Age and growth stage may moderate the degree of internationalization [15], and given that some of the privately owned firms enjoyed a perfect overseas subsidiaries to total subsidiaries score, our findings appear to support this. As some of them only have one subsidiary that happens to be the foreign subsidiary, which suggests they are likely at an early stage of the internationalization process. When compared to the more established firms with a more extended history of internationalization, who may already have shifted focus in other directions or regions, it may be that the Belt and Road initiative provides the appropriate platforms for newly internationalizing firms to expand their foreign operations. Furthermore, with the overarching goal of the belt and road initiative being integration in the stated areas of policy, trade, infrastructure, finance, and people [1], it is possible that these efforts are already bearing fruits and the internationalization process along the Belt and Road countries may have become easier owing to the commitment of the various national governments to see it succeed.

Also, there is *prima facie* evidence that Alibaba's overseas subsidiaries may not be revenue-generating. A possible explanation would be that they serve a strategic rather than commercial purpose, e.g., as research and development centres or as administrative hubs. It seems a logical explanation given the primacy of logistics in the Alibaba business model. Consequently, a thorough study of these subsidiaries may provide additional insights.

Following from above, since firm governance influences international behavior [88] and there are different governance regimes that affect State-owned firms [66], while strict corporate governance requirements negate state ownership [89]; therefore, because our sample consists of listed firms, both private and State-owned firms would be subject to the same

governance requirements. In any case, Estrin et al. [8] posit that listed State-owned enterprises are, in fact, Hybrid firms, and they may be closer in outlook to privately owned firms than to non-listed State-owned enterprises. Future research could compare the internationalization of listed and non listed SOEs.

Lastly, this study is only the first attempt to compute the degree of internationalization of Chinese firms in the context of the belt and road initiative, and while an attempt has been made to utilize as robust a measure as possible, there are still avenues to improve the methodological approach. For instance, whenever data availability permits, using the complex number developed by Fisch and Oesterle [14] may provide an alternative quantitative measure that allows meaningful comparisons to be made.

## Conclusion

Given the scope and magnitude of the belt and road initiative and the unique context it presents for research into the internationalization of Chinese firms, this study attempted to compute the degree of internationalization of Chinese multinationals along the belt and road initiative countries. Using a composite measure, a multi-item index, based on Sullivan [19], Chinese multinationals who had initiated transactions in Belt and Road countries in the period 2013–2018 were compiled from the global investment tracker database. A bottom-up approach was preferred as the transactions were first identified before unmasking the companies responsible for the transactions.

Even though there was no "right" way to measure the degree of internationalization found in the literature, a multi-item indexed was preferred over a single measure of the degree of internationalization given the superiority of multi-item measures of the degree of internationalization. The index used, comprised of four items, FSTS, OSTS, PDIO and GD, which measured the ratio of foreign sales to total sales, overseas subsidiaries to total subsidiaries, psychic dispersion and geographic dispersion respectively. It was hypothesized that state-owned enterprises would rank higher than privately owned enterprises and that psychic distance would influence State-owned enterprises and privately owned firms equally through an identical level of psychic dispersion within the Belt and Road initiative countries. However, both *H1* and *H2* were unsupported. Privately owned enterprises indeed had a higher average $DOI_{BRI}$ than the State-owned enterprises, while the State-owned enterprises scored higher on the psychic dispersion measure, indicating, on average, more psychically diverse operations. This result suggests that perhaps the internationalization of Chinese firms has matured to such a stage that the effects of ownership structure Vis a Vis state ownership are approaching insignificance. This corporate maturity would imply that one of the main explanations often cited as the reason for the unique nature of the internationalization of Chinese firms may have dissipated in relevance. Chinese firms may not now be that much different from their western counterparts; consequently, extant theories may do a better job of explaining their behavior. This notion ought to be tested in future research. Although it should be added that the determinants of Chinese investments in Belt and Road countries differ to their determinants in non-Belt and Road countries [84].

Despite the limitations of this study, it contributes to the literature on the internationalization of Chinese firms as well as the belt and road initiative. It is a first attempt at assessing the Belt and Road initiative and the degree of internationalization of firms. A high degree of internationalization along the Belt and Road initiative countries would suggest the acceptance and adoption of the policy by multinational firms, although, with the caveat that some of the investments and projects may have been planned earlier. Prima facie, the Belt and Road Initiative, as a policy seems to have spurred internationalization equally between State-owned enterprises and privately owned firms.

## Supporting information

**S1 Appendix. List of firms with their respective scores for all variables.**
(DOCX)

**S2 Appendix. List of BRI countries where firms have invested and the corresponding number of Psychic Zones.**
(DOCX)

## Acknowledgments

Comments and suggestions from the anonymous reviewers are gratefully acknowledged.

## Author Contributions

**Conceptualization:** Olawoyin Gregory Adedigba, Runhui Lin.

**Data curation:** Nizam Ud Din.

**Formal analysis:** Olawoyin Gregory Adedigba, Nizam Ud Din.

**Investigation:** Olawoyin Gregory Adedigba.

**Methodology:** Olawoyin Gregory Adedigba, Nizam Ud Din.

**Resources:** Runhui Lin.

**Software:** Nizam Ud Din.

**Supervision:** Runhui Lin.

**Writing – original draft:** Olawoyin Gregory Adedigba.

**Writing – review & editing:** Olawoyin Gregory Adedigba.

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
