## [Decision Letter · Decision Letter 0]

4 Mar 2020

PONE-D-20-03433

The degree of internationalization of Chinese Multinationals along the belt and road initiative countries

PLOS ONE

Dear Mr. Adedigba,

Thank you for submitting your manuscript to PLOS ONE. After careful consideration, we feel that it has merit but does not fully meet PLOS ONE’s publication criteria as it currently stands. Therefore, we invite you to submit a revised version of the manuscript that addresses the points raised during the review process.

The paper is interesting. However, there are important parts to make clear and complete and numerous writting mistakes. Also there are too numerous acronyms must be recalled regularly. Please answer carefully to the numerous remarks of the reviewers.

We would appreciate receiving your revised manuscript by Apr 18 2020 11:59PM. To enhance the reproducibility of your results, we recommend that if applicable you deposit your laboratory protocols in protocols.io, where a protocol can be assigned its own identifier (DOI) such that it can be cited independently in the future. For instructions see: http://journals.plos.org/plosone/s/submission-guidelines#loc-laboratory-protocols

We look forward to receiving your revised manuscript.

Kind regards,

Celine Rozenblat

Academic Editor

PLOS ONE

Journal Requirements:

Reviewers' comments:

Reviewer's Responses to Questions

**Comments to the Author**

1. Is the manuscript technically sound, and do the data support the conclusions?

Reviewer #1: Partly

Reviewer #2: Yes

2. Has the statistical analysis been performed appropriately and rigorously? 

Reviewer #1: No

Reviewer #2: Yes

3. Have the authors made all data underlying the findings in their manuscript fully available?

Reviewer #1: Yes

Reviewer #2: Yes

4. Is the manuscript presented in an intelligible fashion and written in standard English?

Reviewer #1: No

Reviewer #2: No

5. Review Comments to the Author

Reviewer #1: This paper claims private firms have higher degree of internalization rather than state owned Chinese MNCs along the Belt and Road countries. The topic is novel and the conclusion is interesting. Within the particular definition of BRI countries in this paper, the methodology is sufficient to allow the reproduced experiment. However, several unsolved issues still exist in this paper:

1. Literature

Previous literature on the ownership of Chinese MNCs along BRI countries can be included, such as 10.13529/j.cnki.enterprise.economy.2018.11.002, which discusses the discrimination toward state owned capital of SOEs in BRI area.

2. Data & Methodology

1) Actually, the B&R initiative does not have a clear definition of its boundary yet. What are the specific area/countries included in the paper?

2) The numbers and sizes of SOEs and POEs are not the same, so they may not be comparative without any distinction.

3) Similarly, in table 3，only using the number of countries/firms to describe the PDIO and GD is not sufficient, at least the volume of countries/firms should be considered.

4) In table 3, DOI is simple multiplication of 4 items. The four items may not be equally effective in DOI, and then the different coefficients should be considered.

5) In table 3, what is the difference between foreign sales and overseas sales in the indicator FSTS?

3. Writing

Several spelling and grammar mistakes make some paragraphs ambiguous: 1) “However, dspite its economic and politiacal significance of the BRI, there are only a few empirical studies on the internationalization activities of Chinese MNCs in countries along the” in p9. 2) “…because the quantitative studies have mainly examined the causal relationship between the BRI and changes in the levels of Chinese OFDI, a preference for the Difference in differences thechnique for data analysis [4,5,8] was observed.” …

Above all, this paper is unsuitable for publication in its present form. If the above questions can be well solved, this paper probably can be published.

Reviewer #2: The research presents an interesting and important subject in the complex area of internationalization strategies of BRI countries. It attempts to assess the degree of internationalization (DOI) of Chinese firms along the Belt and Road initiative countries and then to find the relation of firm ownership to DOI. But, current manuscript needs some corrections to present the research process and results clearly. Please see the below comments.

- Abstract: it is better to explain the variables of analysis and then the complete results of the research rather than just indicating the first hypothesis.

- Introduction:

• For the reference 9, authors referred to the conditions of internationalization (citing Dunning), however as it is the first time in the manuscript it is necessary to explain these conditions. It is explained on the page 6. The same is for all abbreviations which are introduced for the first time in the manuscript. This will improve the readability of the context.

• on page 3, in contrast to the abstract, results should not be explained in the introduction part. Rather than giving unnecessary information, it is better to explain the case study (BRI project), the problem statement, the importance of doing research in this case study and on this subject, main questions and aims of the research, hypothesizes and the procedural framework of the research.

• The statement of “ Lending credence to the view that under certain conditions, the behaviours of SOEs and POEs in terms of international activities converge and become indistinguishable [7,17]. Suggesting that perhaps along the BRI, the advantages bestowed by state ownership are irrelevant to successful internationalization” is a critical and interesting statement in this research and in better to be deeply explained in the proper part of the manuscript (in the part of results and not introduction).

- Theoretical Framework

• For the first part: the previous studies on BRI considering OFDI trends, different economic activities and economic evolution related to implementation of BRI, are reviewed. However these explanations do not provide new insights for the theoretical framework of this research. They can be helpful to justify the novelty and necessity of this research toward the previous ones.

• In the second part, authors attempted to review the theories of firm internationalization. However instead of first defining the concept and its different determinants, it discussed the internationalization of Chinese firms providing fragmented critiques from different approaches without making clear what exactly they are. (e.g. discussing OLI approach for CFI without explaining Dunning’s approach). This trend is continued in the methodological part and even in the conclusion. So it is strongly recommended to reorganize the theoretical part; first define the concept of “firm internationalization” based on different theories, and clarify its similarities and differences from different conceptual lenses.

Also rather than considering the Dunning’s theory (OLI), or Rugman’s (for FSA, CSA) and Haier and Galanz’s study occasionally, just start with the theories of macroeconomic and then go to the details of firm-base perspectives such as the Uppsala model, the eclectic paradigm, network theories and etc. and shape the theoretical implications of “firm-internationalization”. And now discuss the way that Chinese firms are internationalized. The last paragraph again discusses the difference of this study with the previous studies on BRI, just based on its firm-level scale of the analysis. It has the potential to clarify its contribution to the theories of firm internationalization as well.

• In the third part, authors tried to discuss the FI measurement. Again considering the Uppsala model and methods introduced by Sullivan and Ramawamy et al. however references 14 and 15 are unrelated to the previous methods of measurement. But the point is that this part should be integrated either with the conceptualization part or with the methodological part for building the indexes of analysis.

• The fourth part: hypothesizes, can be explained in the problem statement.

- Methodological Framework

• The last paragraph on page 12, defining the international firm, again have to be transferred to the theoretical part.

• In the last paragraph of the “variables”, on page 15, the three dimensions of structure, attitude and performance are discussed but the GD and its relation to the main determinants of IF is neglected.

• Table 3 has to include the exact formula and method of calculation of each variable and the way to discount the scale differences of these non-weight combination indexes.

• How the variables of GD and PDIO are build and how do they differ from each other.

- Results

• Before presenting the results, it is better to explain the descriptive features of the variables in two groups of firms. For example, in each group, private or public, for each indexes, which firms have the higher and lower values. What are their characteristics, their maturity, the number of subsidiaries, the share of foreign investments, the economic activities (service, production, infrastructure, etc.). And in which BRI countries these POEs and SOEs have the most subsidiaries, trades and why.

• In this part the authors just discussed the result of first hypothesis and the second one is neglected.

This manuscript contains lots of errors in its writing. some are colored in yellow in the attached file.

6. PLOS authors have the option to publish the peer review history of their article (what does this mean?). If published, this will include your full peer review and any attached files.

Reviewer #1: No

Reviewer #2: No

---

## [Author Response · Author response to Decision Letter 0]

18 Apr 2020

1. Response to Academic Editor.

“The paper is interesting. However, there are important parts to make clear and complete and numerous writting mistakes. Also there are too numerous acronyms must be recalled regularly. Please answer carefully to the numerous remarks of the reviewers.”

We appreciate the positive assessment of our work. The paper has been revised and reformatted. Furthermore, care has been taken, to as much as possible, do away with acronyms without affecting the paper. For instance acronyms have been significantly culled from the introduction, Discussion and Conclusion sections, as well as in the main text (e.g. P.21, line 447; P.28, line 571,575; P.30, line 614). In addition in the main text the acronym “MNCs” has been replaced with the word “multinationals”, only retaining it in tables and keywords. Similarly acronyms have been removed from the abstract.

2. Response to Reviewer #1: 

“This paper claims private firms have higher degree of internalization rather than state owned Chinese MNCs along the Belt and Road countries. The topic is novel and the conclusion is interesting. Within the particular definition of BRI countries in this paper, the methodology is sufficient to allow the reproduced experiment. However, several unsolved issues still exist in this paper:”

Literature

“Previous literature on the ownership of Chinese MNCs along BRI countries can be included, such as 10.13529/j.cnki.enterprise.economy.2018.11.002, which discusses the discrimination toward state owned capital of SOEs in BRI area.”

Thank you for this suggestion and providing a fresh perspective.

Data & Methodology

 “Actually, the B&R initiative does not have a clear definition of its boundary yet. What are the specific area/countries included in the paper?”

Thank you for pointing this out. The countries included are based on the official Belt and Road website (http://yidaiyilu.com.gov.cn) as at the time of writing the paper in December 2018). We have included a citation in the first paragraph of the introduction to clarify this (Profiles -Belt and Road Portal, 2018) (P.2; line 46), as well as in the methodology section (P.22, line 453)

“The numbers and sizes of SOEs and POEs are not the same, so they may not be comparative without any distinction.”

You are correct indeed, and we acknowledge that the number and sizes are not the same. However the variable of concern is ownership. Explicitly, whether the firm is owned by the state or not. This is a categorical variable and not subject to firm size. In addition, we do note in our discussion the distinction that state owned firms tend towards the huge infrastructural projects relative to their private counterparts. Furthermore the literature reports mixed findings on the relationship between size and degree of internationalization (Pangarkar, 2008). 

“Similarly, in table 3，only using the number of countries/firms to describe the PDIO and GD is not sufficient, at least the volume of countries/firms should be considered”

We follow the literature in measuring geographic dispersion as a ratio of the potential number of countries in which a firm can have operations and the actual number of countries in which the firm has operations(Ietto-Gillies, 1998). This in this case corresponds to all countries in the belt and road initiative. Furthermore, Ietto-Gilles notes 

“There might be a degree of arbitrariness in the choice of this number;

however, such a choice does not affect the results since the exact value of the

denominator in the index is not very relevant. What is needed is a datum which

expresses the potential number of countries and which can be kept constant for all companies, at any given period of time.” (Ietto-Gillies, 1998, p. 26)

Similarly in adopting the same approach for the measure of psychic dispersion, we conform to the literature as both Sullivan(1994) based on psychic zones as well as Fisch and Oesterle(2003) measured cultural and psychic dispersion in like manner.

We would also like to point out, since both PDIO and GD are dispersion measures, they are designed to capture the extensity of activities and not the intensity. In other words the breadth and scope of activities and not magnitude.

“In table 3, what is the difference between foreign sales and overseas sales in the indicator FSTS?”

Thank you for this observation. It was meant to read overseas subsidiaries. The error has now been fixed.(p.13)

 Writing

“Several spelling and grammar mistakes make some paragraphs ambiguous: 1) “However, dspite its economic and politiacal significance of the BRI, there are only a few empirical studies on the internationalization activities of Chinese MNCs in countries along the” in p9. 

“…because the quantitative studies have mainly examined the causal relationship between the BRI and changes in the levels of Chinese OFDI, a preference for the Difference in differences thechnique for data analysis [4,5,8] was observed.”

Thank you for this observation, we acknowledge the writing errors. All the errors have been corrected.

 “In table 3, DOI is simple multiplication of 4 items. The four items may not be equally effective in DOI, and then the different coefficients should be considered.”

We follow Sullivan (1994, 1996) in using a composite index that is based on a linear addition of the different variables measuring various dimensions of internationalization.. Furthermore, equal weights are assigned to each item because the core thrust of our argument is that the different aspects of internationalization are given adequate consideration. Sullivan (1996) addresses the merits of using this approach rather than varying the weights assigned to each item.

3. Response to Reviewer #2: 

“The research presents an interesting and important subject in the complex area of internationalization strategies of BRI countries. It attempts to assess the degree of internationalization (DOI) of Chinese firms along the Belt and Road initiative countries and then to find the relation of firm ownership to DOI. But, current manuscript needs some corrections to present the research process and results clearly. Please see the below comments”

Abstract: 

“it is better to explain the variables of analysis and then the complete results of the research rather than just indicating the first hypothesis.”

Thank you for this suggestion. The abstract has been rewritten to include a mention of variables and dimensions of internationalization measured as suggested, and the key findings.

Introduction:

 “For the reference 9, authors referred to the conditions of internationalization (citing Dunning), however as it is the first time in the manuscript it is necessary to explain these conditions. It is explained on the page 6. The same is for all abbreviations which are introduced for the first time in the manuscript. This will improve the readability of the context.”

This has been rectified via a short description of the antecedents of internationalization (P.3, line 75-80)

 “on page 3, in contrast to the abstract, results should not be explained in the introduction part. Rather than giving unnecessary information, it is better to explain the case study (BRI project), the problem statement, the importance of doing research in this case study and on this subject, main questions and aims of the research, hypothesizes and the procedural framework of the research.”

 The background of the BRI was provided in the introduction along with the problem statement, significance of the study as well as the aim of the research in the last paragraph on page 3 as well as the first paragraphs of page 4 (p.3 lines 58,67; p.4 lines 80,92; P.5 line 98). Also, page 5 contains the literature review on belt and road initiative research. 

“The statement of “Lending credence to the view that under certain conditions, the behaviours of SOEs and POEs in terms of international activities converge and become indistinguishable [7,17]. Suggesting that perhaps along the BRI, the advantages bestowed by state ownership are irrelevant to successful internationalization” is a critical and interesting statement in this research and in better to be deeply explained in the proper part of the manuscript (in the part of results and not introduction)”.

Thank you for this comment and observation. This excerpt has now been moved to the discussion section (p.28, line 566)

Theoretical Framework

“For the first part: the previous studies on BRI considering OFDI trends, different economic activities and economic evolution related to implementation of BRI, are reviewed. However these explanations do not provide new insights for the theoretical framework of this research. They can be helpful to justify the novelty and necessity of this research toward the previous ones”

You are indeed correct; the aim of the review was to show the gap in the current belt and road research. We identify the missing firm level perspective in extant BRI research, as Internationalization per se has not been investigated, consequently the potential of the BRI to extend internationalization research is highlighted,(p.7 line 142).

“In the second part, authors attempted to review the theories of firm internationalization. However instead of first defining the concept and its different determinants, it discussed the internationalization of Chinese firms providing fragmented critiques from different approaches without making clear what exactly they are. (e.g. discussing OLI approach for CFI without explaining Dunning’s approach). This trend is continued in the methodological part and even in the conclusion. So it is strongly recommended to reorganize the theoretical part; first define the concept of “firm internationalization” based on different theories, and clarify its similarities and differences from different conceptual lenses”.

We appreciate this suggestion. These recommendations have now been incorporated into the paper via paragraphs 1-3 on page 8.

“Also rather than considering the Dunning’s theory (OLI), or Rugman’s (for FSA, CSA) and Haier and Galanz’s study occasionally, just start with the theories of macroeconomic and then go to the details of firm-base perspectives such as the Uppsala model, the eclectic paradigm, network theories and etc. and shape the theoretical implications of “firm-internationalization”. And now discuss the way that Chinese firms are internationalized. “

We appreciate and acknowledge your observation and it has been included in the paper, (P.8 lines 164,175; p.9,lines 188,196; P.10 line 222).Thank you.

“This manuscript contains lots of errors in its writing. some are colored in yellow in the attached file.”

These have all been corrected.

Methodological Framework

 “The last paragraph on page 12, defining the international firm, again have to be transferred to the theoretical part.”

Thank you for this observation, it has now been implemented.

“The last paragraph again discusses the difference of this study with the previous studies on BRI, just based on its firm-level scale of the analysis. It has the potential to clarify its contribution to the theories of firm internationalization as well. In the third part, authors tried to discuss the FI measurement. Again considering the Uppsala model and methods introduced by Sullivan and Ramawamy et al. however references 14 and 15 are unrelated to the previous methods of measurement. But the point is that this part should be integrated either with the conceptualization part or with the methodological part for building the indexes of analysis. The fourth part: hypothesizes, can be explained in the problem statement.”

We appreciate your suggestions and they have been incorporated into the paper. The third part has been done away with and integrated into both the introduction and the methodology sections. Regarding the Hypotheses we are of the opinion that retaining it in the final section of literature review improves the flow and readability of the article. Consequently it was included there.

“In the last paragraph of the “variables”, on page 15, the three dimensions of structure, attitude and performance are discussed but the GD and its relation to the main determinants of IF is neglected”.

Thank you for this observation. The GD has now been added to the discussion.

 “Table 3 has to include the exact formula and method of calculation of each variable and the way to discount the scale differences of these non-weight combination indexes.”

“How the variables of GD and PDIO are build and how do they differ from each other.”

Table 3 has now been updated with the calculation methods.(p.22)

Results

 “Before presenting the results, it is better to explain the descriptive features of the variables in two groups of firms. For example, in each group, private or public, for each indexes, which firms have the higher and lower values. What are their characteristics, their maturity, the number of subsidiaries, the share of foreign investments, the economic activities (service, production, infrastructure, etc.). And in which BRI countries these POEs and SOEs have the most subsidiaries, trades and why.”

“In this part the authors just discussed the result of first hypothesis and the second one is neglected.”

Thank you for this suggestion. A presentation of the descriptive statistics has now been included. As well as two tables displaying the statistics for the individual variables for the state owned enterprises and private owned enterprises respectively (p.23; p.24). In addition, a table with all the firms and their respective scores has been included in the appendix.

---

## [Decision Letter · Decision Letter 1]

12 May 2020

PONE-D-20-03433R1

The degree of internationalization of Chinese Multinationals along the belt and road initiative countries

PLOS ONE

Dear Mr. Adedigba,

Thank you for submitting your manuscript to PLOS ONE. After careful consideration, we feel that it has merit but does not fully meet PLOS ONE’s publication criteria as it currently stands. Therefore, we invite you to submit a revised version of the manuscript that addresses the points raised during the review process.

The paper is now acceptable, but it stlll needs some improvements. Please follow the suggestion of the reviewer 2.

We would appreciate receiving your revised manuscript by Jun 26 2020 11:59PM. To enhance the reproducibility of your results, we recommend that if applicable you deposit your laboratory protocols in protocols.io, where a protocol can be assigned its own identifier (DOI) such that it can be cited independently in the future. For instructions see: http://journals.plos.org/plosone/s/submission-guidelines#loc-laboratory-protocols

We look forward to receiving your revised manuscript.

Kind regards,

Celine Rozenblat

Academic Editor

PLOS ONE

Additional Editor Comments (if provided):

The paper is now acceptable, but it stlll needs some improvements. Please follow the suggestion of the reviewer 2.

Reviewers' comments:

Reviewer's Responses to Questions

**Comments to the Author**

1. If the authors have adequately addressed your comments raised in a previous round of review and you feel that this manuscript is now acceptable for publication, you may indicate that here to bypass the “Comments to the Author” section, enter your conflict of interest statement in the “Confidential to Editor” section, and submit your "Accept" recommendation.

Reviewer #1: All comments have been addressed

Reviewer #2: All comments have been addressed

2. Is the manuscript technically sound, and do the data support the conclusions?

Reviewer #1: Yes

Reviewer #2: Yes

3. Has the statistical analysis been performed appropriately and rigorously? 

Reviewer #1: Yes

Reviewer #2: Yes

4. Have the authors made all data underlying the findings in their manuscript fully available?

Reviewer #1: Yes

Reviewer #2: No

5. Is the manuscript presented in an intelligible fashion and written in standard English?

Reviewer #1: Yes

Reviewer #2: Yes

6. Review Comments to the Author

Reviewer #1: The questions are answered. The ambiguous definitions and data sources are clarified. And the mistakes of written English are corrected properly. This paper offers a novel and interesting conclusion on BRI area .

Reviewer #2: Authors have addressed the comments precisely.

I would like to suggest that rather than giving descriptive explanations in the paragraphs, use more figures, graphs and charts. table 4 & 5 can be integrated. It is worth giving information about the countries that these POEs, SOEs have the most subsidiaries, trades etc. in, and to relate this with the categories of the countries in table 2.

7. PLOS authors have the option to publish the peer review history of their article (what does this mean?). If published, this will include your full peer review and any attached files.

Reviewer #1: No

Reviewer #2: Yes: Maedeh Hedayatifard

---

## [Author Response · Author response to Decision Letter 1]

18 May 2020

1. Response to Academic Editor.

“The paper is now acceptable, but it still needs some improvements. Please follow the suggestion of the reviewer 2.”

Thank you for the positive review of our work. We have taken the additional suggestions to hear and have made the necessary adjustments.

2. Response to Reviewer #2: 

Reviewer #2: Authors have addressed the comments precisely.

“I would like to suggest that rather than giving descriptive explanations in the paragraphs, use more figures, graphs and charts. table 4 & 5 can be integrated. It is worth giving information about the countries that these POEs, SOEs have the most subsidiaries, trades etc. in, and to relate this with the categories of the countries in table 2”.

Figures and Tables

“I would like to suggest that rather than giving descriptive explanations in the paragraphs, use more figures, graphs and charts Table 4 & 5 can be integrated”

We appreciate your suggestion and have presented the sectoral composition of the sample (Fig 1,P.25 line 498) as well as the sector breakdown for both privately and State-owned firms respectively (Figs 2, and 3, P.25 lines 499 and 500) in charts.

Regarding, Tables 4 and 5 , they are presented as such because the discussion of results is done with a delineation of SOEs and POEs as suggested by you in the last round of revisions. Also, a key theme in this paper is the comparison between the two types of ownership. Consequently, we are of the view that your previous suggestion of delineating results of SOEs and POEs suits the paper better. However, to accommodate your suggestion, we have moved Table 5 right next to table 4(P.24 Line 478) for ease of reading. Furthermore, table 6 presents the combined overall statistics for both POEs and SOEs. 

Additional information

“It is worth giving information about the countries that these POEs, SOEs have the most subsidiaries, trades etc”

Thank you for this suggestion. We have now included an appendix (S2) which provides the list of the countries in which the companies have made investments, as well as the corresponding number of psychic zones according to the Ronen and Shenkar classification

---

## [Decision Letter · Decision Letter 2]

13 Jul 2020

The degree of internationalization of Chinese Multinationals along the belt and road initiative countries

PONE-D-20-03433R2

Dear Dr. Adedigba,

We’re pleased to inform you that your manuscript has been judged scientifically suitable for publication and will be formally accepted for publication once it meets all outstanding technical requirements.

Kind regards,

Celine Rozenblat

Academic Editor

PLOS ONE

Additional Editor Comments (optional):

It remains some minor remarks tat would be easy to address.

Reviewers' comments:

Reviewer's Responses to Questions

**Comments to the Author**

1. If the authors have adequately addressed your comments raised in a previous round of review and you feel that this manuscript is now acceptable for publication, you may indicate that here to bypass the “Comments to the Author” section, enter your conflict of interest statement in the “Confidential to Editor” section, and submit your "Accept" recommendation.

Reviewer #1: All comments have been addressed

Reviewer #2: All comments have been addressed

2. Is the manuscript technically sound, and do the data support the conclusions?

Reviewer #1: Yes

Reviewer #2: Yes

3. Has the statistical analysis been performed appropriately and rigorously? 

Reviewer #1: Yes

Reviewer #2: Yes

4. Have the authors made all data underlying the findings in their manuscript fully available?

Reviewer #1: Yes

Reviewer #2: Yes

5. Is the manuscript presented in an intelligible fashion and written in standard English?

Reviewer #1: Yes

Reviewer #2: Yes

6. Review Comments to the Author

Reviewer #1: Authors have addressed comments comprehensively. This article is now acceptable. The added tables and appendix give more detailed information of the data and methods. This article provides the quantitative studies from the perspective of BRI enterprises. The results show that private firms reach higher degree of internationalization and state-owned enterprises show a more psychically dispersed character, which is quite interesting and thought-provoking. Perhaps further quantitative studies will go deeper to explore what organizational, institutional or other salient factors cause this phenomenon and how they effect.

Reviewer #2: The authors have attempted to address the suggestions precisely. some small improvements in line with the previous suggestions are still needed for the publication.

- descriptive information is presented in figure 1,2,3 . It is still suggested to integrate fig 2&3. by integration it was/is meant to keep both information, but just present them in one format of table or figure. this can reduce the number of tables and figures and make the comparison more readable. It is the same for table 4&5 in the previous review rounds.

-In line 538, the sentence is incomplete.

-Authors were encouraged to give information about the countries that these POEs and SOEs have "the most" subsidiaries and trades in. So it is not necessary to include the raw list of "all countries" for "all firms" in the Appendix 2. it is sufficient to add this information in the descriptions and results of the part "the psychic and geographic dispersion."

7. PLOS authors have the option to publish the peer review history of their article (what does this mean?). If published, this will include your full peer review and any attached files.

Reviewer #1: No

Reviewer #2: No

---

## [Editor Report · Acceptance letter]

16 Jul 2020

PONE-D-20-03433R2 

The degree of internationalization of Chinese Multinationals along the belt and road initiative countries 

Dear Dr. Adedigba:

I'm pleased to inform you that your manuscript has been deemed suitable for publication in PLOS ONE. Congratulations! Your manuscript is now with our production department. 

Kind regards, 

on behalf of

Prof. Celine Rozenblat 

Academic Editor

PLOS ONE